# A randomized controlled trial on anonymizing reviewers to each other in peer review discussions

**Charvi Rastogi[1], Xiangchen Song[1], Zhijing Jin[2], Ivan Stelmakh[3], Hal Daumé III[4], Kun Zhang[5], Nihar B. Shah** [1] *

**1** Machine Learning Department, Carnegie Mellon University, Pittsburgh, Pennsylvania, United States of America, **2** Computer Science Department, ETH Zürich, Zürich, Switzerland, **3** New Economic School, Moscow, Russia, **4** Department of Computer Science, University of Maryland, College Park, Maryland, United States of America, **5** Philosophy Department, Carnegie Mellon University, Pittsburgh, Pennsylvania, United States of America

* nihars@cs.cmu.edu

**Data Availability Statement:** In our work, we analyze data obtained from the scientific conference UAI (Conference on Uncertainty in

## Abstract

Many peer-review processes involve reviewers submitting their independent reviews, followed by a discussion between the reviewers of each paper. A common question among policymakers is whether the reviewers of a paper should be anonymous to each other during the discussion. We shed light on this question by conducting a randomized controlled trial at the Conference on Uncertainty in Artificial Intelligence (UAI) 2022 conference where reviewer discussions were conducted over a typed forum. We randomly split the reviewers and papers into two conditions–one with anonymous discussions and the other with non-anonymous discussions. We also conduct an anonymous survey of all reviewers to understand their experience and opinions. We compare the two conditions in terms of the amount of discussion, influence of seniority on the final decisions, politeness, reviewers' self-reported experiences and preferences. Overall, this experiment finds small, significant differences favoring the anonymous discussion setup based on the evaluation criteria considered in this work.

## 1 Introduction

Peer review serves as the backbone of scientific research, underscoring the importance of designing the peer-review process in an evidence-based fashion. The goal of this work is to conduct a carefully designed experiment in the peer-review process to enhance our understanding of the dynamics of discussion among reviewers. Our experiment is conducted in the peer-review process of a Computer Science conference. We note that conferences in Computer Science are typically ranked at par or higher than journals, review full-length papers, and are frequently a terminal publication venue.

In many publication venues, pertinently in conference-based venues in the field of Computer Science, reviewers of a paper first provide their independent reviews, and then engage in

Artificial Intelligence) 2022 for publication of research in AI. For this, we worked in collaboration with the conference co-organizers of UAI 2022 (Kun Zhang). We are not able to share this data in any form, because of the following reasons: 1. This is highly sensitive data concerning authors and reviewers both, where authors and reviewers provide their honest opinion with the belief that this would not affect them personally. 2. is a double-blind conference and maintaining the double-blindness is an important aspect of the reviewing process. Releasing de-identified data would dilute the double-blindness. 3. It would violate the confidentiality agreement between the researchers who participated in the conference and the conference organizers. We are only able to share the aggregated statistics, which are provided in the manuscript. Further, any data requests may be sent to the Association for Uncertainty in Artificial Intelligence that is the institutional body responsible for running the conference UAI 2022, or to the Institutional Review Board of Carnegie Mellon University (contact via irb-review@andrew. cmu.edu).

**Funding:** This work was supported by the following grants awarded to Nihar Shah: Office of Naval Research (ONR) grant N000142212181 (https:// www.nre.navy.mil/media/document/funding-opportunity-n00014-22-s-b001), National Science Foundation (NSF) grant 1942124 (https://www.nsf. gov/awardsearch/showAward?AWD_ID=1942124) and NSF grant 2200410 (https://www.nsf.gov/ awardsearch/showAward?AWD_ID=2200410). Kun Zhang acknowledges support by NSF Grant 2229881 (https://www.nsf.gov/awardsearch/ showAward?AWD_ID=2229881) and by the National Institutes of Health (NIH) under Contract R01HL159805 (https://reporter.nih.gov/search/ NnBR6YW-CEyGMKrv_Pzy0w/project-details/ 10705824). The funders had no role in study design, data collection and analysis, decision to publish, or preparation of the manuscript. There was no additional external funding received for this study.

**Competing interests:** NO authors have competing interests.

a discussion about the paper with the other reviewers of that paper. The discussions take place on a typed forum where reviewers type in their opinions and responses to other reviewers asynchronously. We specifically target an important aspect of this discussion process: *anonymity of reviewers to other reviewers in the discussion*. It is important to note that in our setting, reviewers of papers are always anonymous to their authors and vice versa, commonly referred to as "double-anonymous" or "double-blind" peer review, and we do not study or intervene in that aspect of the peer-review process.

The ongoing discourse within the academic community regarding the anonymization of reviewers to each other during discussions has spanned several years. Despite anecdotal arguments from stakeholders on both sides of the debate, we do not have an evidence-based understanding of the actual effects of anonymity in discussions. Anonymizing discussions carries a set of potential advantages and drawbacks. For instance, it is often hypothesized that anonymizing discussions helps alleviate biases associated with reviewer identities such as more senior or famous reviewers' opinions dominating. This is echoed by research on group discussions [1–3] which finds that participants sometimes use other participants' social status as a heuristic for assessing the credibility of their opinions. Another way this bias can manifest is through reduced participation of junior reviewers in discussion. In line with this [4], reported increased comfort in students when participating in anonymous online discussion forums like Piazza. Additionally, it has been suggested that anonymous discussions can mitigate fraud wherein one reviewer (with an undisclosed conflict of interest with the authors of the paper being reviewed), reveals the identity of the other reviewers to the authors of the paper, to ultimately help coerce the reviewers into accepting the paper [5]. Conversely, proponents of revealing reviewer identities argue that it fosters a deeper comprehension of reviews and perspectives based on knowledge of the review writer's background. Additionally, program chairs have expressed concerns that anonymity may diminish politeness in discussions. Related research [6–8] studying the role of identity in group discussions finds that anonymity can lead to unregulated behaviour. In the past, the peer-review processes in different venues have taken different approaches to the setup for reviewer discussions. For example, among conferences in the field of machine learning and artificial intelligence, AAAI 2021, IJCAI 2022 and other conferences enforced anonymous reviewer discussions whereas many other conferences such as ICML 2016 and FAccT 2023 did not.

In addition to impacting policy design in conference peer review, understanding the impacts of anonymizing discussions among reviewers has broader implications towards understanding the role of participants' identities in group discussion dynamics and outcomes. For instance, it is also relevant for research funding agencies that allocate grants annually through panel-based discussions. Although group discussions involving experts are commonly assumed to improve final decision quality compared to individual decision-makers, controlled experiments studying panel discussions in peer review, have found that group discussions in fact lead to a significant increase in the inconsistency of decisions [9–11]. It is an open problem to understand the underlying causes [12, 13], and this work contributes to a better understanding of discussion dynamics by shedding light on the role of participants' identities in the discussion process by conducting a randomized controlled trial.

With this motivation, we conduct a study on the review process of a top-tier publication venue in the research field of artificial intelligence: The 2022 Conference on Uncertainty in Artificial Intelligence (UAI 2022). The study takes two principal approaches to quantify the advantages and disadvantages of enforcing anonymity between reviewers during the discussion phase. First, we design and conduct a randomized controlled trial to examine the impact of enforcing anonymity on reviewer engagement, reviewer politeness and the influence of seniority on final decisions. Second, we conduct a systematic survey-based study which offers

a more reliable foundation than anecdotal evidence and hypotheses towards informing policy decisions. Our anonymous survey provides an understanding of statistical trends in reviewers' preferences and their experiences in the discussion phase.

## 2 Related work

Certain aspects of peer-review discussions among reviewers have been studied before to understand the dynamics of the discussion phase. Three studies [9–11] performed controlled experiments in the peer-review process for grant proposals to examine the reliability of the (non-anonymous) group discussion phase in the decision-making. These studies compare the agreement between reviewers' independent initial reviews with the agreement between multiple panels' decisions after discussion among members from the same panel. They find that the level of agreement *across* panels decreases in comparison with the agreement across individuals before discussion. A similar study was conducted by [14] in the peer review of hospital quality, yielding a similar conclusion, "*discussion between reviewers does not improve reliability of peer review.*"

To understand the sociological phenomena at play in peer-review discussions [12], conducted a controlled study that exposed reviewers to manufactured scores from other (fictitious) reviewers. In their study, reviewers updated their scores 47% of the time. Along the dimension of gender, women reviewers updated their scores 13% more frequently than men, and more so when they worked in male-dominated fields. Initially high scores were lowered 64% of the time, as opposed to initially low scores which were raised only 24% of the time. [13] conducted a randomized controlled trial in conference peer-review to test for herding behaviour in the reviewer discussion phase. Here, herding behaviour indicates participants' tendency to follow the opinion of the first person to speak-up, shown to be present in other social situations [15]. Interestingly, the outcomes of [13] do not suggest any presence of herding in peer-review discussions. In a large survey of participants of the International Conference on Software Engineering spanning years 2014–2016 [16], find that in their preferences, respondents are split nearly equally on the question of having anonymity between co-reviewers.

Outside of peer review, prior research in sociology examines the effect of status in group discussions and decision-making, where status is based on task-irrelevant but socially valued characteristics of the discussants. In situations where participants' identities are shown [1–3], found that participants may choose to second or reject others' opinions based on their status or perceived credibility. Specifically addressing anonymity in social discussions, a study by [4] on student participation in online discussion boards such as Piazza reveals that students are more comfortable to engage in discussions when they are anonymous. On the other hand [8], find that in anonymous settings, the weakness of social-context cues tends to make people's behaviour relatively self-centered and unregulated, unconcerned about making a good appearance. Group behavior becomes more extreme, more impulsive, and less socially differentiated [6, 7]. However [17], note that computer-mediated discussion (even anonymized) can attenuate social cues and lead to equalization phenomena.

Studying the role of reviewers' identities in the discussion phase and its outcomes is somewhat analogous to the oft-studied question on the effect of revealing authors' identities on peer-review outcomes. This topic has been extensively debated in the scientific community. Amidst these opinions, rigorous experiments and analyses have paved the way towards a scientific approach to addressing this question [18–25] (see [26, Section 'Biases pertaining to author identities'] for a survey). Analogously in this work, we rigorously measure the effects of revealing reviewers' identities to each other, on the discussion phase.

## 3 Experiment setting and design

### 3.1 Setting of the experiment

The experiment was conducted in the peer review process of the 2022 edition of the Uncertainty in Artificial Intelligence conference (UAI 2022). Our study delved into the specifics of the "discussion phase" in the peer review process of UAI 2022, that took place after the initial reviews were submitted, as shown in Fig 1. In this phase reviewers and authors communicated asynchronously via a typed forum, to discuss their respective opinions of the paper. The discussion phase for each paper began after the deadline for initial reviews and ended concurrently with the deadline for final paper decisions. Each paper's discussion involved typically three to four reviewers who were assigned to review that paper and one meta-reviewer (equivalent to the associate editor for journal publication), alongside the authors of the paper. This discussion took place on an online interface (OpenReview.net) which takes in typed posts from participants. For the papers assigned to them, each reviewer is expected to carefully read the reviews written by the other reviewers as well as the authors' responses, and participate in the discussion. During the discussion phase, the reviewers had the option to update their review, which could include updating their review score, to reflect any change of opinion. The determination of the final acceptance of the paper was then done by the meta reviewer based on the initial reviews, the authors' responses, these discussions, and potentially the meta reviewer's own perceptions of the paper.

The UAI 2022 conference was double-anonymous in its conduct, wherein authors were not shown the identities of their reviewers, and vice versa. Within the discussion interface, the reviewers' identities were visible to their corresponding meta-reviewers, while the meta-reviewers' identities were concealed from the reviewers. Notably, our study introduced a key change where the visibility of fellow reviewers' identities was contingent on their assigned condition, as described in the next paragraph.

### 3.2 Design of the experiment

We designed a randomized controlled trial to investigate the effect of anonymity among reviewers in the discussion phase. Each submitted paper and each participating reviewer were assigned at random to one of the two conditions: the "non-anonymous" condition, where reviewers were shown the identities of other reviewers assigned to the same paper, and the "anonymous" condition where fellow reviewers' identities were not accessible. The assignment to conditions was uniformly random for both papers and reviewers. To prevent potential spill-over effects, reviewers within each condition were then matched with papers in the same condition (This also helps address fraud concerns. For instance, AAAI 2020 showed every reviewer just a random fraction of the submitted papers during bidding to mitigate fraud). All

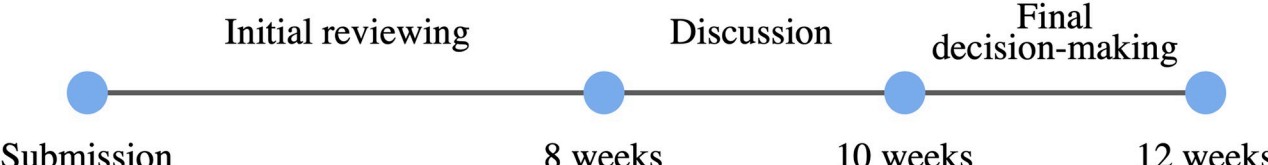

**Fig 1. Timeline of the peer-review process of typical machine learning and artificial intelligence conferences.** Upon the release of initial reviews, authors of papers and the reviewers have an asynchronous discussion. Finally, meta-reviewers aggregate the results of the review process into final decisions. The duration of each stage varies across conferences, and this figure corresponds to the UAI 2022 review process with the duration of each stage rounded to weeks.

reviewers were informed at the beginning of the review process that a randomized controlled trial would be conducted to compare differently anonymized reviewing models. To mitigate the Hawthorne effect, specifics of the experiment were withheld (It is a common practice to inform reviewers that they are part of an experiment without disclosing the specifics. For instance, in [27], reviewers were informed that the research focused on examining the NIH review process, and they would be evaluating modified versions of actual R01 proposals. However, the nature of these modifications was intentionally left undisclosed to the reviewers). Importantly, reviewers were informed that only aggregated statistics would be publicly disclosed, and they were given the choice to opt out of the experiment. At the onset of the discussion phase, reviewers were apprised of whether their discussions would be anonymous or not. We note that the meta-reviewers could see the reviewers' identities in both the conditions and the meta-reviewer pool was common across the two conditions. All meta-reviewers were asked to address reviewers in a way that kept their identities confidential in all discussions.

In conjunction with the randomized controlled trial, we conducted a survey to provide a holistic understanding of the pros and cons of anonymity in the discussion phase. Administered to all reviewers and meta-reviewers, the survey sought to capture valuable insights into their experiences and perspectives. Specifically, the survey gathered reviewers' feedback related to the discussion type they were assigned to, in order to uncover the impact of anonymity (or lack thereof) on their experience in the review process. Furthermore, the survey included questions probing into the aspects of the discussion phase reviewers prioritise. Lastly, we sought reviewers' personal preferences regarding anonymous versus non-anonymous discussions. We elaborate on the specifics of the survey in Section 4.

### 3.3 Details of the experiment

We now delve into the particulars of how the experiment unfolded. Reviewers for the UAI 2022 conference were recruited (independent of the experiment) from December 21, 2021 until February 16, 2022. Prior to commencement of the experiment, all reviewers for the conference were sent an email on April 29, 2022 explaining them the experiment and with an option to opt out. The conference received 701 paper submissions out of which 351 were assigned to the anonymous condition and the remaining 350 to the non-anonymous condition. Over the course of the review process, 69 papers were withdrawn (most papers were withdrawn after reviews were released), leaving 322 in the anonymous condition and 310 in the non-anonymous condition.

At the beginning of the review process 581 reviewers had signed up for reviewing, out of which 65 reviewers dropped out (some reviewers did not submit their reviews in time), resulting in 263 reviewers in the anonymous condition and 253 reviewers in the non-anonymous condition. To substitute for the reviewers that dropped out, meta-reviewers and program chairs added emergency reviewers, such that each new reviewer was added to papers from only one of the two conditions. Following this, the conference had a total of 289 reviewers in each condition. The outcomes of the peer review process of UAI 2022 saw a total of 230 submissions accepted corresponding to an acceptance rate of 36.4% (Please note that the acceptance rate reported in this paper is different from what was announced at the conference, since the conference used the original 712 complete submissions as the denominator, while we consider only those 632 papers that were not eventually withdrawn). Out of these, 116 accepted papers belonged to the anonymous condition and 114 papers belonged to the non-anonymous condition, giving an acceptance rate of 36.0% and 36.7% in the respective conditions. Fisher's exact test for difference in the acceptance rate between the two conditions does not indicate any significant difference ($p$-value = 0.86).

**Table 1. Some preliminary statistics of interest measured across the two experiment conditions in the discussion phase of UAI 2022.** Statistics in the bottom four rows are used towards RQ1.

| Statistic of interest | Anonymous | Non-anonymous |
|---|---|---|
| Number of papers | 322 | 310 |
| Number of reviewers | 289 | 289 |
| Paper acceptance rate | 0.36 (116 out of 322) | 0.37 (114 out of 310) |
| Number of senior reviewers | 169 | 181 |
| Number of {paper-reviewer} pairs | 1163 | 1118 |
| Number of discussion posts written by reviewers | 611 | 514 |
| Number of {paper-reviewer} pairs with posts | 419 | 385 |
| Number of {paper-reviewer} pairs without posts | 744 | 733 |

### 3.4 Reviewer seniority information

This study explores the indirect impact of reviewer seniority during the discussion phase. To achieve this, we gathered data pertaining to the seniority of all reviewers. Specifically, we retrieved reviewers' current professional status from their profiles on Openreview.net. In cases where this information was unavailable, we conducted manual searches on the Internet to find their professional status. Using this information, we established a binary indicator of seniority: post-doctoral researchers and graduate researchers were categorized as junior researchers, while professors and other professional researchers such as research scientists and research engineers were categorized as senior researchers. This classification covered all participating reviewers, and resulted in 181 seniors out of 289 reviewers in the non-anonymous condition and 169 seniors out of 289 reviewers in the anonymous condition, as detailed in Table 1.

### 3.5 Ethics statement

The experiment was reviewed and approved the Institutional Review Board of Carnegie Mellon University. The participants' consent was not obtained as the data were analyzed anonymously.

## 4 Main analyses

In this section, we address research questions on different aspects and impacts of anonymity in reviewer discussions in peer review.

### 4.1 RQ1: Do reviewers discuss more in the anonymous or non-anonymous condition?

Recall that the research question asks whether there is a difference in the amount of participation by reviewers in the discussion phase, between the anonymous and non-anonymous condition. To answer this question, we performed the following analysis. In both conditions, for each reviewer-paper pair, we first computed the number of discussion posts made by that reviewer for that paper (excluding the initial review by the reviewer). We then computed the mean number of discussion posts across all reviewer-paper pairs in each condition. If the difference in means between the two conditions is significantly larger than zero in magnitude, we conclude that anonymity among reviewers (or lack thereof) in the discussion phase has an effect on the amount of discussion participation observed.

 **Results.** Table 1 provides the relevant statistics to address RQ1. We see that the mean number of posts made by reviewers during the discussion phase in the anonymous condition

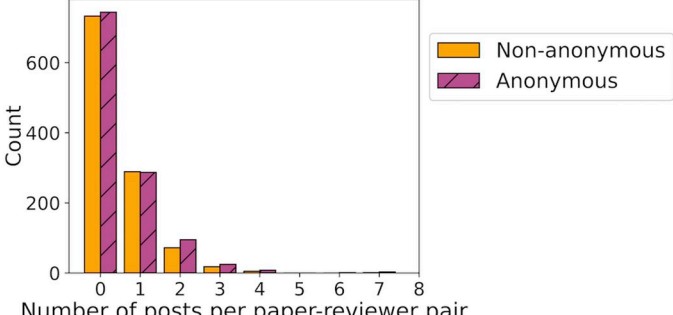

**Fig 2. Visualization of the distribution of count of discussion posts written by a reviewer for a paper in the anonymous and non-anonymous condition in UAI 2022.** The x-axis shows the number of posts made for by a reviewer for a paper.

0.53 (611 out of 1163) is higher than that in the non-anonymous condition 0.46 (514 out of 1118). To evaluate the significance of this difference we conducted a permutation test with 1, 000, 000 iterations, which gave a two-sided p-value of 0.051. Further, we have visualised the distribution of number of posts made corresponding to each paper-reviewer pair in Fig 2.

Given the low rates of posting overall in both the conditions, as evidenced by the frequency of zero in Fig 2, we considered a binary indicator variable indicating post or no post. Precisely, for each paper-reviewer pair this indicator was set to one if there existed any posts associated with the pair and zero otherwise. We then tested for significance of difference across the two conditions using Fisher's exact test. The outcome contingency table is depicted in the last two rows of Table 1 and the corresponding two-tailed p-value is 0.43. This implies no significant difference between the two conditions in posting vs not posting.

We delved further into the data, stratifying posting behavior by seniority of reviewers. On analysing the engagement metrics separately for reviewers based on their seniority, we observed that junior reviewers posted on average 0.58 (273 out of 468) times when anonymous, and 0.57 (232 out of 410) times when non-anonymous, while senior reviewers posted 0.49 (338 out of 695) times when anonymous and 0.40 (282 out of 708) times when non-anonymous.

Next, we looked at the distribution across seniority level within the paper-reviewer pairs where atleast one post was made by the reviewer for the paper. Therein, in the non-anonymous condition, senior reviewers made 214 of the total 385 paper-reviewer pairs, which is approximately 56%. Similarly, in the anonymous condition, senior reviewers made 230 of the total 419 paper reviewer pairs, which is approximately 55%. Thus, within paper-reviewer pairs with posts, we see that senior reviewers participated more than junior reviewers, however their participation was similar across the two conditions.

Given some potential concerns around suppression of participation of junior reviewers in non-anonymous settings [4], these results are surprising and suggest a need for further inquiry.

## 4.2 RQ2: Does seniority have a higher influence on final decisions when non-anonymous than anonymous?

The objective was to test for the presence of any additional influence of senior reviewers towards the final decisions when the identities of fellow reviewers are visible. Consequently, this analysis restricted attention to a subset of papers which had at least one senior reviewer

**Table 2. Statistics of interest measured in the experiment conditions for research question RQ2 on the influence of seniority on final decisions.** These statistics are based on a subset of the papers in UAI 2022 which have at least one senior reviewer and one junior reviewer.

| Statistic of interest | Anonymous | Non-anonymous | *p*-value |
|---|---|---|---|
| Number of papers accepted | 89 (out of 242) | 94 (out of 242) | 0.71 |
| Paper decision closest to senior reviewers, $\beta_p = 1$ | 96 (out of 242) | 122 (out of 242) | |
| Paper decision closest to junior reviewers, $\beta_p = -1$ | 70 (out of 242) | 60 (out of 242) | |
| Mean of $\beta_p$ | 0.11 | 0.26 | 0.04 |

and one junior reviewer. Let $\mathcal{P}_a$ be the set of all such papers in the anonymous condition and $\mathcal{P}_{\tilde{a}}$ be the set of such papers in the non-anonymous condition. For every such paper, our analysis centered on the seniority of the reviewers whose scores, prior to the discussion phase, were closest to the final decision made after discussions. Accordingly, for any paper $p$, we defined $C_p$ as the reviewer or the set of reviewers whose pre-discussion scores were closest to the final decision, as follows. If paper $p$ was accepted, then $C_p$ is the reviewer (or set of reviewers in case of a tie) who gave the highest pre-discussion score to paper $p$ among all its reviewers; if paper $p$ was rejected, then $C_p$ is the reviewer or set of reviewers who assigned it the lowest score before the discussion. Next, for any paper $p$, we introduced a scalar $\beta_p$ defined as follows: $\beta_p = 1$ if $C_p$ includes at least one senior reviewer but no junior reviewer, $\beta_p = 0$ if $C_p$ includes both senior and junior reviewers, and $\beta_p = -1$ if $C_p$ includes only junior reviewers.

To compare the influence of seniority on final decisions across the two conditions, we considered the following test statistic which measures the difference across the two conditions of the extra influence of senior reviewers on the final decision compared to junior reviewers,

$$T_2 = \frac{\sum_{p \in \mathcal{P}_a} \beta_p}{|\mathcal{P}_a|} - \frac{\sum_{p \in \mathcal{P}_{\tilde{a}}} \beta_p}{|\mathcal{P}_{\tilde{a}}|}. \tag{1}$$

Here, a statistically significant value of $T_2$ would indicate that senior reviewers exhibit distinct influence on final decisions depending on whether the reviewers are anonymous or non-anonymous to each other.

**Result.** The statistics related to this analysis are provided in Table 2. First, as detailed in Row 1, there was no statistically significant difference in the acceptance rate of papers in the two conditions, with an acceptance rate of 37% in the anonymous condition and 39% in the non-anonymous condition. The difference in acceptance rates between the conditions has a Fisher-exact *p*-value of 0.71. Now, in reference to the research question posed, we observe that the final decision was closest to a senior reviewer's initial score 50% of the times in the non-anonymous condition and 40% of the times in the anonymous condition. Meanwhile, respectively, the final decision agreed most with a junior reviewer 25% times and 30% times, thus indicating a clear disparity in the seniority of the reviewer whose score was apparently most closely followed for the final decision. The last row of Table 2 shows the mean value of the scalar $\beta_p$ across all papers. We tested for statistical significance of the test statistic $T_2$ in (1) using the permutation test method with 1,000,000 iterations, which yielded a two-sided *p*-value of 0.04.

## 4.3 RQ3: Are reviewers more polite in the non-anonymous condition?

In tackling this research question, we focused on the text of the discussion posts written by reviewers following their initial review. Our aim was to assess the politeness of the discussions posted in the two conditions to investigate any potential disparities therein.

First, we assigned a politeness score to the raw text of each of the discussion posts written by reviewers. We considered the range of the politeness scores to be 1 (highly impolite) to 5 (highly polite). Recent work [28] demonstrated the successful use of commercial large language models (LLMs) such as ChatGPT for rating politeness of scientific review texts via careful prompting. They validated the accuracy and consistency of their method in several ways, such as comparing the generated outputs against human annotation. In our work, we adopted a similar technique of prompting LLMs to score a given text on politeness. However, to protect the privacy of discussion posts in UAI 2022, we avoided commercial APIs that record users' queries. Instead, we locally deploy an open-sourced LLM, Vicuna 13B [29], which achieves close to state-of-the-art performance [29, 30].

Following common prompt engineering practices [31, 32], we instructed the model with the task: "We are scoring reviews based on their politeness on a scale of 1–5, where 1 is highly impolite and 5 is highly polite." Using the few-shot prompting method, we included three scored texts in the prompt corresponding to different politeness scores. These examples were sourced from [33]. The exact text of the prompt is provided in S1 Appendix. Further, to mitigate bias due to the ordering of the few-shot examples, we created six paraphrased versions of the prompt by varying the order of the examples. Since the output generated by LLMs can be different across iterations, each version is queried ten times, and the mean across all paraphrases and iterations yields the final politeness score. For texts exceeding the LLM token limit, we took equal-sized non-overlapping sub-parts of the text such that each subpart satisfies the token limit and the total number of sub-parts is minimized. The politeness score of the larger text was obtained by averaging the scores of its sub-parts.

The process resulted in a politeness score for each discussion post ranging from 1 to 5. To validate the consistency of the generated scores, we measured the correlation of scores generated for the same prompt across iterations, over 1125 unique prompts. We found a significant correlation between two randomly picked iterations for each prompt query, with a Pearson correlation coefficient of 0.42 with a two-sided p-value smaller than 0.001. This p-value indicates the likelihood that a dataset with no correlation could produce a Pearson correlation as extreme as the observed value of 0.42.

**Results.** As noted in the last row of Table 1, there were 611 discussion posts made by reviewers in the anonymous condition and 514 in the non-anonymous condition. Fig 3 visualizes the distribution of the politeness scores obtained for the posts in the two conditions. We observed that posts in the anonymous and the non-anonymous condition received similar scores with mean politeness scores of 3.73 and 3.71 respectively.

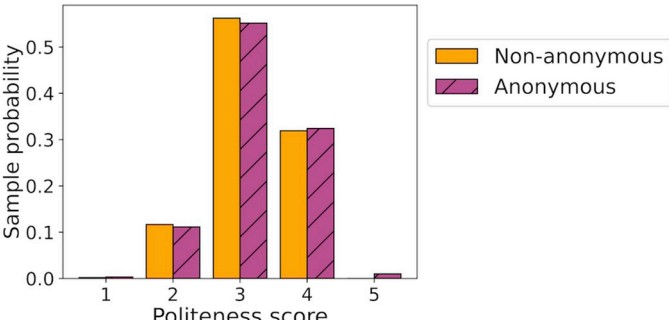

**Fig 3. Visualization of the distribution of politeness scores obtained for all the discussion posts written by reviewers in the anonymous and non-anonymous condition in UAI 2022.** The height of the left-most bar indicates the sample probability of a score falling in the interval [1,2) and so on for each bar.

In this analysis, it is important to note that our test is based on the politeness scores assigned to the discussion posts that took place in each condition. However, as we saw in Section 4.1 the posting rates differed in the two conditions, and hence the politeness data may reflect selection bias. For instance, it is possible that the average politeness observed in the anonymous condition is high because of the higher level of participation by senior reviewers in comparison to the non-anonymous condition. In Section 4.1 we saw that the rate of posting for senior reviewers was different in the two conditions at 0.49 when anonymous and 0.40 otherwise.

To account for this, we grouped the posts based on the seniority of the reviewer and conducted comparisons within each group. That is, the politeness scores of posts by senior reviewers in one condition were compared against only those by senior reviewers in the other condition and same for posts by junior reviewers. The resulting Mann-Whitney $U$ test [34] for significant difference in the politeness of discussion posts across the two conditions revealed no significant difference. It is noteworthy that even though reviewers could not see each other's identities in the anonymous condition, the meta-reviewers and program chairs could see the identity of the reviewers in both conditions.

The Mann-Whitney test yielded a normalized $U$-statistic of 0.49 with a $p$-value of 0.72. The normalized $U$-statistic approximates the probability in our sample that a randomly chosen score in the non-anonymous condition is higher than a randomly chosen score in the anonymous condition. The mathematical definition of the $U$-statistic is provided alongside details about the test in S2 Appendix.

Across different seniority levels, we see that in the anonymous condition, senior and junior reviewers receive a mean politeness score of 3.68 and 3.79 respectively. Meanwhile, in the non-anonymous condition, senior and junior reviewers received a mean politeness score of 3.66 and 3.77 respectively.

## 4.4 RQ4: Do reviewers' self-reported experiences differ?

Recall that we conducted an anonymous survey for all the UAI reviewers and meta-reviewers to glean insights about their experiences in the discussion phase of UAI 2022. Some questions in the survey were designed to understand differences in reviewers' (self-reported) experiences in the two conditions. Since the pool of meta-reviewers was the same for both the conditions, we excluded them from this part of the survey. Specifically, respondents were asked to provide Likert-style responses based on their personal experience, to the following:

1. I felt comfortable expressing my own opinion in discussions including disagreeing with and commenting on reviews of other reviewers.

2. My opinion was taken seriously by other reviewers.

3. I could understand the opinions and perspectives of other reviewers given the information available to me.

4. Discussions between reviewers were polite and professional.

5. Other reviewers took their job responsibly and put significant effort in participating in discussions.

Survey respondents answered each of the five questions with exactly one of the five options: Strongly disagree, Somewhat disagree, Neither agree nor disagree, Somewhat agree, Strongly agree.

**Table 3. For each survey question, we map the Likert-scale responses as: Strongly disagree → 1, Somewhat disagree → 2, Neither agree nor disagree → 3, Somewhat agree → 4, Strongly agree → 5.** With this mapping, we computed the mean response in the two experiment conditions, displayed in columns 2 and 3. Column 4 provides the effect size, which is the normalized Mann-Whitney *U* statistic in our analysis. This value approximates the sample probability that a randomly chosen response from one condition was higher than a randomly chosen response from the other condition. Using the permutation test defined in (5) with 100,000 iterations, we report here the two-sided p-value of the test for each survey question in the last column. These p-values do not include a multiple testing correction, and are already insignificant.

| Survey question (shortened for illustration) | Mean response (Anonymous) | Mean response (Non-anonymous) | Effect size | *p*–value |
|---|---|---|---|---|
| 1. I felt comfortable expressing my own opinion. | 3.85 | 4.10 | 0.43 | 0.33 |
| 2. My opinion was taken seriously by other reviewers. | 3.56 | 3.64 | 0.48 | 0.77 |
| 3. I could understand the opinions of other reviewers. | 4.03 | 4.06 | 0.47 | 0.96 |
| 4. Discussions between reviewers were professional. | 4.13 | 4.27 | 0.47 | 0.97 |
| 5. Others were responsible & effortful in discussion. | 3.37 | 3.17 | 0.46 | 0.73 |

**Result.** We received responses from 64 reviewers in the non-anonymous condition and 68 reviewers in the anonymous condition. The overall survey response rate was 22.8%. The outcomes of the Mann-Whitney *U* test for each survey question are provided in Table 3, with the effect size and the p-value in the last two columns. Additionally, we visualize the Likert-scale responses for each question in Fig 4. We did not observe any significant difference in survey participants' self-reported experiences across the two conditions. Further, the difference across the conditions was small with the normalized *U*-statistic lying between 0.43 and 0.48, where this value approximates the sample probability that a randomly chosen response from one condition was higher than a randomly chosen response from the other condition. An effect size of 0.5 implies that along the axis mentioned in the corresponding survey question such as politeness respondents had similar experiences in both conditions. More details about the test and the derivation of the p-values are provided in S2 Appendix.

## 4.5 RQ5: Do reviewers prefer one condition over the other?

We surveyed all the reviewers and meta-reviewers asking for their overall preference between the two conditions, framed as follows:

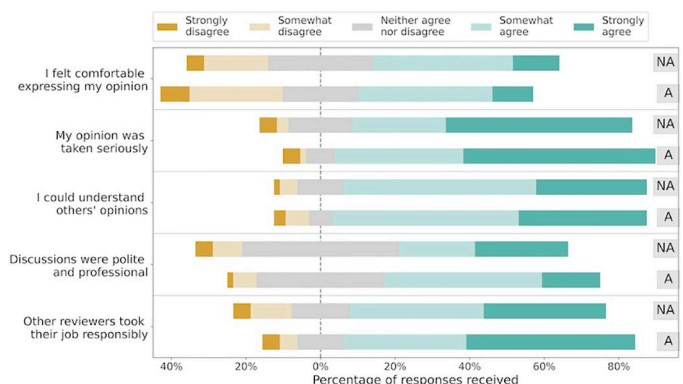

**Fig 4. Survey outcomes on reviewers' self-reported experience in the two conditions.** Respondents provided Likert-style responses for each question which have been visualized here. The markers 'NA' and 'A' indicate responses from the non-anonymous and the anonymous condition respectively. For each survey question, we tested for difference in the participants' responses across the two conditions employing the Mann-Whitney U test. There was no significant difference in participants' experience corresponding to any of the survey questions. More details are provided in Table 3.

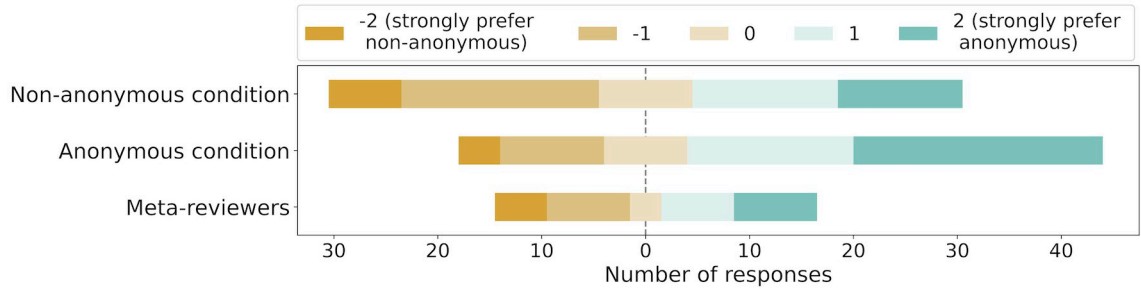

**Fig 5. Survey outcomes on reviewers' and meta-reviewers' reported overall preference between the two conditions in the experiment.** Respondents provided Likert-style responses ranging from 'I strongly prefer reviewer identities to be shown to other reviewers' to 'Indifferent' to 'I strongly prefer reviewer identities to be hidden from other reviewers.'

*Overall, what is your preference on whether reviewer identities should be SHOWN to other reviewers or HIDDEN? This question is about your general opinion and not restricted to your UAI 2022 experience.*

In the responses, participants were provided five options and could choose at most one from these options. We provide the options here along with their numeric mapping in this analysis:

- I strongly prefer reviewer identities to be HIDDEN from other reviewers: 2

- I weakly prefer reviewer identities to be HIDDEN from other reviewers: 1

- Indifferent: 0

- I weakly prefer reviewer identities to be SHOWN to other reviewers: -1

- I strongly prefer reviewer identities to be SHOWN to other reviewers: -2.

**Result.** In total we obtained 159 responses out of which 124 were from reviewers and 35 from meta-reviewers. Under the chosen mapping of responses to numbers, we computed the mean over all responses. This numeric mean corresponding to the responses obtained is 0.35 (Cohen's d = 0.25), suggesting a weak preference for reviewer identities to be anonymous. The responses had a standard deviation of 1.38, giving a 95% confidence interval of [−2.36, 3.08]. Furthermore, we visualize the distribution of the responses in Fig 5, separated based on the group of the reviewers. We see that reviewers in the anonymous condition are skewed in their overall preference in favour of the anonymous condition. Meanwhile, meta-reviewers and reviewers from the non-anonymous condition are relatively more equally distributed across the two overall preferences.

## 4.6 RQ6: What aspects do reviewers consider important in making the policy decision regarding anonymizing reviewers to each other

In the survey, we asked reviewers and meta-reviewers about the aspects they considered important in making conference policy decisions regarding anonymity between reviewers in discussions. Specifically, we asked them to rank the following six aspects according to what they found to be most important to least important:

- Reviewers take their job more responsibly and put more effort in writing reviews and participating in discussions.

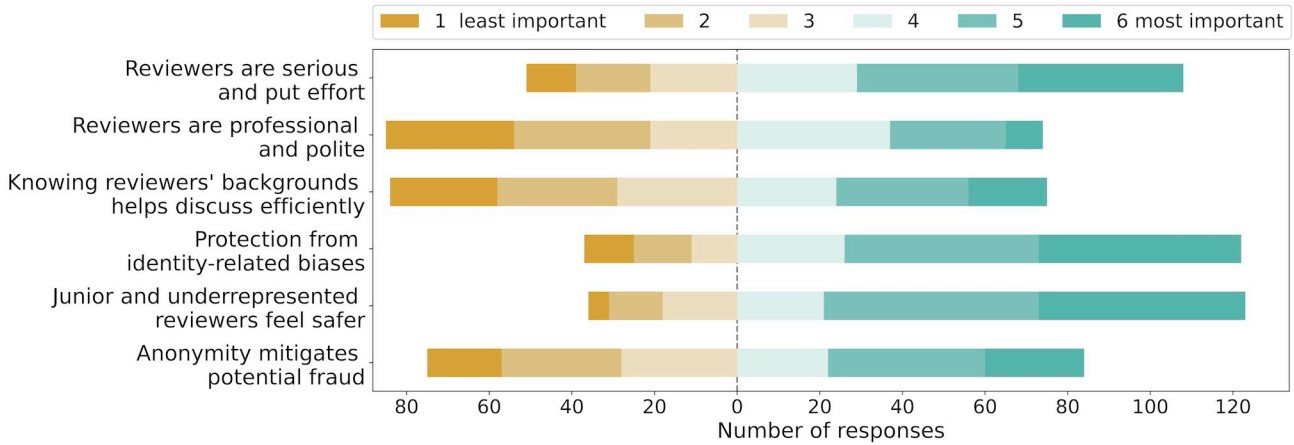

**Fig 6. Survey outcomes on reviewers' ranking of importance of different aspects in making conference policy decisions regarding anonymity between reviewers in discussions.**

- Reviewers communicate with each other in a more polite and professional manner.

- Knowledge of backgrounds of other reviewers helps discuss more efficiently.

- Reviewers are protected from identity-related biases.

- Reviewers (especially junior or from underrepresented communities) feel safer to express their opinions.

- Anonymity helps mitigate potential fraud (e.g., authors pressurizing reviewers with the help of their friend who is one of the reviewers).

Each respondent was asked to provide a rating for each aspect from 1 to 6 where 1 indicated that the aspect was least important to them and 6 indicated the aspect was most important to them.

**Result.** We had a total of 159 respondents answer all of the six importance questions. The outcomes are visualised in Fig 6. As shown, the highest importance was given to the aspect regarding reviewers feeling of safety in expressing their opinion, with a mean importance score of 4.56. Second most important aspect was about protecting reviewers from identity-related biases with a mean importance score of 4.44. The two least important aspects were regarding politeness and professionalism of communication among reviewers and efficiency of discussion with a mean score of 3.16 and 3.4 respectively.

## 4.7 RQ7: Have reviewers experienced dishonest behavior due to reviewer identities being shown to other reviewers?

The survey contained a question concerning reviewers' and meta-reviewers' current and past experiences relating to reviewer anonymity in peer review discussions. Specifically, the survey question stated:

Have you ever experienced any dishonest behavior due to reviewer identities being shown to other reviewers? Examples include: authors colluding with their friend who is a reviewer and attempting to contact other reviewers and pressurize them to accept the paper review-ers contacting other reviewers outside of the review system.

To comprehensively answer the questions, respondents had the option to choose one or more of the following options: "Yes, in UAI 2022," "Yes, in another venue," "Not sure," and "No." Further, the survey asked respondents to describe the dishonest attempt(s) they have experienced or those they may suspect as a free-text response, if they felt comfortable doing so.

**Results.** Out of the 167 respondents, 7% (12 out of 167) mentioned that they have experienced dishonest behaviour relating to anonymity in the discussion phase. Among the 12 yes responders, 8 were reviewers and 4 were meta-reviewers, and 11 chose "Yes, in another venue," while one respondent chose "Yes, in UAI 2022." From the remaining 155 respondents, 14 were unsure and the remaining majority said no.

Six respondents provided free-text responses relating their experiences with dishonest behavior, mainly describing two distinct behaviors. Some respondents mentioned absence of anonymity in the discussion phase resulting in the disclosure of reviewers' identities to the authors of the paper under review, possibly due to an undeclared connection between one of the reviewers and the authors, and on being coaxed to review a certain way. Beyond the inherent issue of compromising double-anonymity between reviewers and authors, the breach of author anonymity has been known to escalate to coercion, with authors pressuring reviewers to provide favorable evaluations of their submissions [5]. Second, two respondents noted that renowned researchers have in the past imposed their identity and stature on other reviewers to increase the influence of their review in the discussion phase, although this behavior may not necessarily be termed as 'dishonest.'

### 4.8 Free-text comments

The survey respondents were also given the option of providing free-text comments under the question,

> **Final comments**. Any additional thoughts on whether reviewer identities should be shown to other reviewers? Do you have any relevant experience to share (in UAI 2022 or elsewhere)?

Out of the provided comments, several were more general comments for the program chairs which were conveyed to the program chairs, but are omitted here as they are not relevant to the topic of this paper. Some other comments repeated their responses to the other questions, and we also omit them. We summarize the remaining relevant comments covering various aspects below:

- **Discussion quality**. One respondent noted that hiding reviewer identities helps focus the discussion on the review content. However, opposing perspectives were shared by other respondents who argued that showing identities incentivizes better-quality review writing, as the reviewer may be appreciated for it by their colleagues. Additionally, transparency in identities contributes to more meaningful and effective discussions by revealing reviewers' backgrounds. It also facilitates transfer of reviewing know-hows from senior to junior reviewers. Lastly, a respondent suggested that non-anonymity may benefit reviewers by fostering more connections in the research community.

- **Dishonest behavior**. Respondents raised concerns about non-anonymity in discussion phase leading to breaking of reviewer-author anonymity in different ways. One concern involves authors who are also acting as reviewers of other papers, and have co-reviewers that are also reviewing the authors' paper. Here, it possible that the authors deduce the identities of their paper's reviewers by comparing writing styles with their co-reviewers whose identity

is visible to them. In another scenario discussed, a reviewer with an undeclared conflict of interest could potentially disclose the other reviewers' identities to the authors, possibly leading to reviewer coercion in favour of the paper, explicitly in the non-anonymous setting and implicitly otherwise. Here the respondent added that implicit coercion via manipulating the discussion itself would be easier in the anonymous condition. To address such concerns a respondent suggested having a policy guaranteeing protection for whistleblowers, in case of reported fraud in peer review.

- **Implementation**. Several respondents emphasized the importance of timely and clear communication, coupled with strict enforcement of anonymity policies in discussions. One respondent cited an instance where, despite having an anonymous discussion setting, a meta-reviewer disclosed a reviewers' identity to others in their exchanges. To potentially have the benefits of both settings, some respondents proposed a policy wherein reviewers' identities are revealed to each other after the conclusion of the discussion phase.

- **Adherence**. Certain respondents pointed out that hostile confrontations can occur in both settings. Interestingly, even in anonymous settings, well-known researchers have been known to reveal their identity to overtly influence the discussion in their favour.

## 5 Discussion

To improve peer review and scientific publishing in a principled manner, it is important to understand the quantitative effects of the policies in place, and design policies in turn based on these quantitative measurements. In this work, we focus on the peer-review policy regarding anonymity of reviewers to each other during the discussion phase.

This work provides crucial evidence, based on data from the experiment in UAI 2022, that there are some potentially adverse effects of removing anonymity between reviewers in paper discussions. Revealing reviewers' identities to each other leads to slightly lower engagement in the discussions and leads to undue influence of senior reviewers on the final decision. Some anecdote-based arguments in favour of showing reviewers' identities have focused on its importance for maintaining politeness of discussions among reviewers. However, in the scope of our experiment, we find that there is no significant difference in the politeness of reviewers' discussions across the two conditions. Notably, 7% of survey respondents reported having witnessed dishonest practices due to non-anonymity among reviewers, which were supported by provided anecdotes.

To conduct a complete investigation of possible impacts of anonymity policies in reviewer discussions, we collect reviewers' perspectives on this issue via anonymous surveys in UAI 2022. While the responses reveal a small difference in participants' preference over the two conditions, they also indicate no significant different in participants' experiences in the two conditions, across dimensions such as comfort, effectiveness and politeness. We encourage the policymakers of peer-review processes to take these findings into account when designing policies, and invite other publication venues to conduct similar experiments.

It is interesting to view our study in the context of previous research on the outcomes of panel-based discussions in grant review. In controlled experiments conducted in non-anonymous settings [9–11], found that group discussions led to a significant increase in the inconsistency of decisions. Synthesizing these findings with our experiment's outcomes suggests that the inconsistency in decisions in non-anonymous settings may stem from biases in each group's decision-making, potentially resulting in divergent outcomes across different groups.

### 5.1 Limitations

We now discuss some limitations of our study. The surveys administered in UAI 2022 were anonymous with a response rate of approximately 20%, which brings the possibility of selection bias in the results. However, it is important to note that our observed response rate aligns reasonably with response rates commonly observed in surveys within the field of Computer Science. For example, surveys by [35] in the NeurIPS 2021 conference saw response rates of 10–25%. [25] conducted multiple surveys: an anonymous survey in the ICML 2021 and EC 2021 conferences had response rates of 16% and 51% respectively; a second, non-anonymous opt-in survey in EC 2021 had a response rate of 55.78%. The opt-in survey by [36] for authors in the peer-review process of AAEE 2011 had a response rate was 28%.

As briefly mentioned in Section 3, reviewers were informed at the beginning of the review process about our experiment regarding anonymity of discussions, but specific details were withheld to mitigate the Hawthorne effect. However, it is possible that reviewers were hesitant to let other reviewers' opinions affect their opinion as that could be undesirably perceived as tied to the knowledge of reviewers' identities. Next, we discussed in Section 4.3 our implementation of LLM-based politeness analysis. Here, it is important to note that given the variability of performance of different LLMs and subjectivity of politeness evaluation, despite the robustness checks conducted in Section 4.3, the results could vary with a different approach to assigning politeness scores to review texts.

Another limitation is our use of a binary grouping of reviewers into seniors and juniors based on their recent professional status. However, this categorization may be overly quantized and hence combine reviewers with a diverse range of expertise. Additionally, our experiment was conducted on the OpenReview.net interface and it is possible that the results could vary when working with a different interface.

To fully assess the effectiveness of any such interventions or policies in the peer-review process, an alternative approach would be to establish clear objectives for the process (e.g., identifying flaws in papers), create submissions with objective ground truth, and evaluate reviews against this benchmark. Along these lines, some past work focuses on identifying flaws in full papers [37–41], [26, Section 'Ensuring Correctness'] and evaluating LLMs' ability to identify flaws in papers and identifying abstracts with superior results [42], [26, Section 'AI Reviewing']. However, implementing this in real peer-review processes poses several challenges. It requires constructing complete research papers and then requires many reviewers to spend time evaluating these papers. While we strongly advocate for more experimentation to enhance the understanding of the peer-review process and inform evidence-based policy adjustments—especially since substantial researcher time is spent on it—the decision to undertake such efforts ultimately rests with organizers and the research community, who must weigh the additional workload against the potential benefits.

## Supporting information

**S1 Appendix.**
(PDF)

**S2 Appendix.**
(PDF)

## Acknowledgments

We thank James Cussens who served as program co-chair (with Kun Zhang), and general co-chairs–Cassio de Campos and Marloes Maathuis of UAI 2022 for their support. We thank the

OpenReview.net team, and particularly Harold Rubio, Melisa Bok and Celeste Martinez Gomez, and Nadia L'Bahy for helping us set up the experiment on the OpenReview.net platform. We also thank Giorgio Piatti for assistance to Zhijing Jin in setting up the LLMs. We are grateful to all the participants of the UAI peer-review process for their time and effort.

## Author Contributions

**Conceptualization:** Ivan Stelmakh, Hal Daumé, III, Kun Zhang, Nihar B. Shah.

**Data curation:** Charvi Rastogi, Xiangchen Song, Ivan Stelmakh, Nihar B. Shah.

**Formal analysis:** Charvi Rastogi, Ivan Stelmakh, Nihar B. Shah.

**Funding acquisition:** Nihar B. Shah.

**Investigation:** Charvi Rastogi, Zhijing Jin, Ivan Stelmakh.

**Methodology:** Charvi Rastogi, Xiangchen Song, Zhijing Jin, Ivan Stelmakh, Hal Daumé, III, Kun Zhang, Nihar B. Shah.

**Project administration:** Nihar B. Shah.

**Resources:** Kun Zhang, Nihar B. Shah.

**Software:** Charvi Rastogi.

**Supervision:** Nihar B. Shah.

**Validation:** Charvi Rastogi, Zhijing Jin.

**Visualization:** Charvi Rastogi, Nihar B. Shah.

**Writing – original draft:** Charvi Rastogi, Nihar B. Shah.

**Writing – review & editing:** Charvi Rastogi, Zhijing Jin, Kun Zhang, Nihar B. Shah.

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
