## [Decision Letter · Decision Letter 0]

25 Jul 2024

PONE-D-24-08652A Randomized Controlled Trial on Anonymizing Reviewers to Each Other in Peer Review DiscussionsPLOS ONE

Dear Dr. Shah,

Thank you for submitting your manuscript to PLOS ONE. After careful consideration, we feel that it has merit but does not fully meet PLOS ONE’s publication criteria as it currently stands. Therefore, we invite you to submit a revised version of the manuscript that addresses the points raised during the review process. Please see the Editor's Comments below along with the comments of two reviewers that must be addressed. 

We look forward to receiving your revised manuscript.

Kind regards,

Henry Hugh Bailey, Ph.D

Academic Editor

PLOS ONE

Journal Requirements:

When submitting your revision, we need you to address these additional requirements. 1. Please ensure that your manuscript meets PLOS ONE's style requirements, including those for file naming. The PLOS ONE style templates can be found at https://journals.plos.org/plosone/s/file?id=wjVg/PLOSOne_formatting_sample_main_body.pdf and https://journals.plos.org/plosone/s/file?id=ba62/PLOSOne_formatting_sample_title_authors_affiliations.pdf 2. Thank you for stating the following in the Acknowledgments Section of your manuscript: "We thank James Cussens who served as program co-chair (with Kun Zhang), and general co-chairs–Cassio de Campos and Marloes Maathuis of UAI 2022 for their support. We thank the OpenReview.net team, and particularly Harold Rubio, Melisa Bok and Celeste Martinez Gomez, and Nadia L’Bahy for helping us set up the experiment on the OpenReview.net platform. We also thank Giorgio Piatti for assistance to Zhijing Jin in setting up the LLMs. We are grateful to all the participants of the UAI peer-review process for their time and effort. This work was supported by ONR N000142212181, NSF 1942124 and NSF 2200410 grants. KZ acknowledges support by NSF Grant 2229881 and by the National Institutes of Health (NIH) under Contract R01HL159805. The experiment was reviewed and approved by the Carnegie Mellon University Institutional Review Board." We note that you have provided funding information that is not currently declared in your Funding Statement. However, funding information should not appear in the Acknowledgments section or other areas of your manuscript. We will only publish funding information present in the Funding Statement section of the online submission form. Please remove any funding-related text from the manuscript and let us know how you would like to update your Funding Statement. Currently, your Funding Statement reads as follows: "This work was supported by the following grants awarded to Nihar Shah: Office of Naval Research (ONR) grant N000142212181 (https://www.nre.navy.mil/media/document/funding-opportunity-n00014-22-s-b001), National Science Foundation (NSF) grant 1942124 (https://www.nsf.gov/awardsearch/showAward?AWD_ID=1942124) and NSF grant 2200410 (https://www.nsf.gov/awardsearch/showAward?AWD_ID=2200410).Kun Zhang acknowledges support by NSF Grant 2229881 (https://www.nsf.gov/awardsearch/showAward?AWD_ID=2229881) and by the National Institutes of Health (NIH) under Contract R01HL159805 (https://reporter.nih.gov/search/NnBR6YW-CEyGMKrv_Pzy0w/project-details/10705824). 
 The funders had no role in study design, data collection and analysis, decision to publish, or preparation of the manuscript. There was no additional external funding received for this study." Please include your amended statements within your cover letter; we will change the online submission form on your behalf.

**Editor's Comments:**As per my e-mail e-mail of June 17th, the abstract needed to be cut back to the word limit, with numbered and bulleted points removed. I did receive your e-mail explaining that this had been done but that you had not received a reply from the Journal regarding submitting the corrected paper.. I was also trying to get the system reopened to allow for the resub, but eventually in the interest of time I sent the paper out for review with the abstract as submitted. I trust that your paper will be resubmitted with the abstract correctly formatted and within the word limit. In addition to this, please pay careful attention to the points raised by two reviewers below. 

Reviewers' comments:

Reviewer's Responses to Questions

**Comments to the Author**

1. Is the manuscript technically sound, and do the data support the conclusions?

Reviewer #1: Partly

Reviewer #2: Yes

2. Has the statistical analysis been performed appropriately and rigorously? 

Reviewer #1: Yes

Reviewer #2: I Don't Know

3. Have the authors made all data underlying the findings in their manuscript fully available?

Reviewer #1: Yes

Reviewer #2: No

4. Is the manuscript presented in an intelligible fashion and written in standard English?

Reviewer #1: Yes

Reviewer #2: Yes

5. Review Comments to the Author

**Reviewer #1: **

This is a very comprehensive analysis for a review process that is not very common, at least when assessing scientific articles and conference abstracts: a review process with internal discussions between the reviewers. I only know this from juries that have to decide on prizes that are beyond scientific. Therefore, in my opinion, the relevance of the research question is not very high. It turns out that the differences between anonymous and non-anonymous discussions are generally small. This is quite a surprising and important finding. However, it does not correspond to the general statement made by the authors (“Overall, the experiment reveals evidence supporting the advantages of an anonymous discussion setup in the peer-review process”). It would be more correct to state that differences were either not noticeable at all or were very small in favor of anonymous discussions.

**Reviewer #2**: 

Firstly - "thank you", this is an interesting topic that ought to interest all researchers / reviewers / editors (effectively a global audience, beyond UAI). That said, for those unfamiliar with the multi-stage decision making that underpins this survey, it might be helpful to have a graphical representation of the study design (a la CONSORT).

What is not clear, is the determination of acceptance - the point at which a paper is included and specifically, how its "score" is computed. Do reviewers trade/haggle/negotiate/interact as a result of intermediate typed comments? How is the endpoint/consensus determined ? The reason for asking this, is that downstream you report the extent to which reviewers converge with final scores and this mechanism does not seem to be included in the descriptive account of the study. Crucially, the reader needs to know how the final score is computed.

Out of interest, did any of the authors take on the role of reviewers - and if so, then what were their NA / A status?

Although the primary focus of the study relates to the issue of reviewer anonymity, this is intimately connected to that of reviewer seniority and this strand of investigation is not consistently followed in all reported analysis.

Figure 1 reveals a key issue. The modal number of discussion posts is zero, with a marginally higher proportion in the A group. This type of behavioural data invites us to consider what constitutes the most appropriate measure of "average". Is it meaningful to report an average response of 0.53 given the overwhelming majority of nil posts?

Once we remove the nil response group, it seems that there is parity for category 1, with a trend towards greater number of interactions in the A group vs NA group. Surely this is a 2-step issue, what is the initial response - (post or not post), then consider what is going on amongst those who DO post. It'd be so more informative if we knew whether there is any difference in post/no-post rates by seniority. Is it feasible to redesign Figure 1 to show this? Perhaps with a data table?

The text states that a permutation test with 1,000,000 iterations was used to test for significance - this just feels like an overkill, especially given the resulting (marginal) p-value. There are other ways of assessing concordance in ordinal data of this type - might a contingency table be more informative?

This issue of "closest" fit to acceptance score .... might it be informative to take each pair and compute the differences for the senior & junior reviewers, code the winner / loser as 1/0 respectively and then run a sign test?

I completely LOVE the politeness index section - however, it does introduce a massive confounder in that you/we all have to "believe" in ChatGBT. Personally I think it's novel but risky and wouldn't want to go to the wall in its defence just now. You might want deal with this in the discussion? To be honest, I could see this being taken out of this current manuscript to form the basis of a 2nd separate paper. Just a thought.

When categorical response data are "scored" (as in #4.5) and then treated as cardinal, being represented by a MEAN (with 2 decimal places) then I do wonder what is going on. The reader is entitled to see at least, a descriptive account of the raw data. So, in this case it would improve matters if we have a revised form of Table 3 that shows the frequency of response category for each of the 5 statements, separately for N/A and A groups. We can then visualise the distribution for each statement and extent of any similarity between statements. Of course, it'd be simple to reslice the salami and checkout the patterns for senior/junior reviewers.

The "scoring" system imposes the researchers ex post value on responses. Hence the statement "I strongly prefer reviewer identities to be HIDDEN from other reviewers" (value=>2) reported by a single reviewer, entirely negates the statement "

"I weakly prefer reviewer identities to be SHOWN to other reviewers" (value =>-1). Please review this aspect of the analysis and consider alternative ways to handle these data?

I prefer the reporting format in Figures 3 & 4 which deal with a somewhat similar data type.

It might be useful in the Discussion section to consider whether any aspects of the design might have been modified, as well assessing the value of any findings of note, for example in guiding best practice in conference abstract reviews. One suggestion that I would have considered is the use of some well-designed abstracts with specific methodological/data reporting "flaws" to test the extent to which such material is differentially noted; additionally, if UAI conference abstract design so permits, how about infiltrating (hidden) identification of authors by loading the abstract with references to their papers?

6. PLOS authors have the option to publish the peer review history of their article (what does this mean?). If published, this will include your full peer review and any attached files.

Reviewer #1: No

Reviewer #2: **Yes: **Paul KIND

---

## [Author Response · Author response to Decision Letter 0]

10 Sep 2024

Please see the attached file on response to reviewers. Thank you very much for all your time and effort in reviewing our paper and for providing such valuable feedback!

---

## [Editor Report · Decision Letter 1]

24 Sep 2024

PONE-D-24-08652R1A Randomized Controlled Trial on Anonymizing Reviewers to Each Other in Peer Review Discussions

PLOS ONE

Dear Dr. Shah,

I have considered your revised manuscript and I am satisfied that you have effectively taken care of the issues raised by the reviewers. 

I would like to suggest two minor changes before moving forward.

1-    The  first point raised by Reviewer 1 included this statement:

*This is a very comprehensive analysis for a review process that is not very common, at least when assessing scientific articles and conference abstracts: a review process with internal discussions between the reviewers. I only know this from juries that have to decide on prizes that are beyond scientific. Therefore, in my opinion, the relevance of the research question is not very high. *

…and your response to this was:

*We would like to clarify that conferences in fields related to artificial intelligence, which review many tens of thousands of papers every year, typically do have such discussions. We would also like to clarify that these conferences review full papers (not just abstracts) and are frequently the terminal venues of publications. In addition, discussions between reviewers also takes place in various grant panel reviews (although these occur in face-to-face settings rather than typed forums). *

Like Reviewer 1, in my field this is not standard practice. I would suggest that you include a statement in your Introduction about full papers being reviewed at AI conferences, for the benefit of readers from different fields.

2-    Some of your methods are written in the present tense. It may improve the article if you change this and use the past-tense whenever you are describing/explaining your work. Introductory content that describes current thinking etc should stay in the present tense, but I would suggest that when you are describing your study design, survey, analysis etc that you standardize to the past tense. 

We look forward to receiving your revised manuscript.

Kind regards,

Henry Hugh Bailey, Ph.D

Academic Editor

PLOS ONE
---

## [Author Response · Author response to Decision Letter 1]

19 Nov 2024

Thank you very much for your feedback on the manuscript, and your suggestions which we greatly appreciate! 

Regarding "AI conferences": We mention this distinction in our manuscript in the Introduction section as footnote 1. The text is reproduced here: 

‘Conferences in Computer Science are typically ranked at par or higher than journals, review full-length papers, and are considered to be a terminal publication venue.’

Regarding tense: Thank you for your comment, we have incorporated it in our manuscript and changed the tense accordingly. 

(By the way, the "update file order" in the editorialmanager was not working)

---

## [Editor Report · Decision Letter 2]

29 Nov 2024

A Randomized Controlled Trial on Anonymizing Reviewers to Each Other in Peer Review Discussions

PONE-D-24-08652R2

Dear Dr. Shah,

We are pleased to inform you that your manuscript has been judged scientifically suitable for publication and will be formally accepted for publication once it meets all outstanding technical requirements.

Kind regards,

Henry Hugh Bailey, Ph.D

Academic Editor

PLOS ONE

---

## [Editor Report · Acceptance letter]

10 Dec 2024

PONE-D-24-08652R2 

PLOS ONE

Dear Dr. Shah, 

I'm pleased to inform you that your manuscript has been deemed suitable for publication in PLOS ONE. Congratulations! Your manuscript is now being handed over to our production team.

Kind regards, 

on behalf of

Dr. Henry Hugh Bailey 

Academic Editor

PLOS ONE